# The Interaction of the Endogenous Hydrogen Sulfide and Oxytocin Systems in Fluid Regulation and the Cardiovascular System

**DOI:** 10.3390/antiox9080748

**Published:** 2020-08-14

**Authors:** Nicole Denoix, Oscar McCook, Sarah Ecker, Rui Wang, Christiane Waller, Peter Radermacher, Tamara Merz

**Affiliations:** 1Clinic for Psychosomatic Medicine and Psychotherapy, Ulm University Medical Center, 89081 Ulm, Germany; nicole.denoix@uni-ulm.de; 2Institute for Anesthesiological Pathophysiology and Process Engineering, Ulm University Medical Center, 89081 Ulm, Germany; sarah.ecker@uni-ulm.de (S.E.); peter.radermacher@uni-ulm.de (P.R.); tamara.merz@uni-ulm.de (T.M.); 3Faculty of Science, York University, Toronto, ON M3J 1P3, Canada; ruiwang@yorku.ca; 4Department of Psychosomatic Medicine and Psychotherapy, Nuremberg General Hospital, Paracelsus Medical University, 90419 Nuremberg, Germany; christiane.waller@klinikum-nuernberg.de

**Keywords:** cystathionine γ-lyase, hydrogen sulfide, oxytocin receptor, psychosomatic medicine, cardiovascular, fluid regulation

## Abstract

The purpose of this review is to explore the parallel roles and interaction of hydrogen sulfide (H_2_S) and oxytocin (OT) in cardiovascular regulation and fluid homeostasis. Their interaction has been recently reported to be relevant during physical and psychological trauma. However, literature reports on H_2_S in physical trauma and OT in psychological trauma are abundant, whereas available information regarding H_2_S in psychological trauma and OT in physical trauma is much more limited. This review summarizes recent direct and indirect evidence of the interaction of the two systems and their convergence in downstream nitric oxide-dependent signaling pathways during various types of trauma, in an effort to better understand biological correlates of psychosomatic interdependencies.

## 1. Introduction

The gasotransmitter hydrogen sulfide (H_2_S) (endogenously produced by cystathionine γ-lyase (CSE), cystathionine β-synthase (CBS) and 3-mercaptopyruvate-sulfurtransferase (3MST) (as depicted in Figure 1)) and the neuroendocrine oxytocin (OT) system have been recently shown to not only possibly play parallel roles in the heart and the brain in response to trauma, but also to influence one another. Trauma can lead to cardiovascular impairments and disease which worsens the outcomes of intensive care patients and increases morbidity and mortality [1,2]. Trauma is either a consequence of a deep emotional pain or a physical injury. Psychological trauma is due to a strong emotional response to a life-threatening event. In contrast, physical trauma can be defined as physiological injury or an impact against the body. Current research underscores the fact that physical and psychological trauma share physiological correlates [3]. OT and H_2_S are relevant in models of both psychological and physical trauma, displaying cardio-protective, anti-oxidative and anti-inflammatory effects [4,5,6,7].

There is a dearth of information regarding the direct interaction between the H_2_S and OT systems: in human myometrial strip biopsies, sodium hydrosulfide (NaHS), a H_2_S releasing salt, decreased the frequency of oxytocin-induced myometrial contractions, mediated by ATP-sensitive potassium channels (K_ATP_-channels) [10]. You et al., reported an inverse correlation between CSE and oxytocin receptor (OTR) expression patterns in the myometrium in pregnant women at term before or after the onset of labor [11]. They also showed that both H_2_S produced locally and exogenously administered NaHS were able to suppress OTR expression and activate phosphatidylinositol 3-kinase (PI3K), extracellular signal-regulated protein kinase (ERK) and K_ATP_-channels in pregnant human myometrial cells [11]. Finally, in a CSE knock out (CSEko) model, Akahoshi et al. were able to show that pregnant mice presenting with preeclampsia-like symptoms displayed dysfunctional contraction of the mammary myoepithelial cells, which was attributed to decreased OTR, and also had impaired uterine contraction responses to OT [12]. The paucity of literature on this subject is a limitation, which led us to investigate parallel roles for the two systems, in the heart and brain, which may suggest their interaction. Specifically, this review explores the literature on both H_2_S and OT systems to help to shed light on their interaction in the regulation of fluid balance, blood volume [4,13], blood pressure [14,15,16,17] and heart rate [18,19].

## 2. The Neuroendocrine Control of Fluid Hemostasis

The hypothalamus is the central integrative structure for the regulation of blood and body fluid volume and osmolality [20]. In response to changes of the peripheral fluid homeostasis, the hypothalamus regulates the blood pressure and heart rate [4]. The supraoptic (SON) and paraventricular nuclei (PVN) are of particular interest with regard to the interaction of the H_2_S and OT systems in the regulation of fluid homeostasis [4]. In trans-ascending aortic constriction rat models of heart failure, chronic selective activation of PVN neurons producing OT was effective in increasing cardiac parasympathetic tone and ameliorating the loss of heart function and myocardial injury [21,22]. H_2_S has been reported to depolarize magnocellular neurons of the PVN in a dose–response fashion, hinting at its possible role in regulating autonomic and endocrine functions [23]. Moreover, in water deprived (24 h) rats, an intracerebroventricular injection of sodium sulfide (Na_2_S), another H_2_S-releasing salt, led to increases plasma concentrations of OT and a decrease of hypothalamic nitrate/nitrite [4]. The authors concluded that H_2_S regulates the roles of both the behavioral and neuroendocrine responses to water deprivation independently of nitric oxide (NO) and stimulates OT secretion by inhibiting the NO system [4]. In a more recent study, in rat hypothalamic explants H_2_S was also shown to be a (positive) regulator of OT in response to acute osmotic stimulus, whereas NO played a key role as a negative neuroendocrine modulator of OT [24]. Additionally, Coletti et al. localized constitutive expression of CBS in the rat hypothalamic SON and PVN, but an important omission in this model which may limit their findings was that they did not look for the interaction of CSE [24]. This omission assumes importance when considering the complex interaction of CSE and CBS, which were found to be expressed in an inverse fashion after cerebral injury [25]. A further word of caution is warranted regarding the translational value of the above experiments [4,22,23,24]: they were all performed in rat models, and some of them on tissue explants. Ex vivo studies in general are a limited representation of the in vivo situation, and rats may not be the most appropriate model organism to study the effects of NO interactions, considering their much higher NO levels. Endogenous NO production in rats is at least an order of magnitude greater than in pigs and humans [26], suggesting that the results here may only be applicable to the rat.

In a clinically relevant porcine hemorrhagic shock study [27], variable expressions of OT, OTR, CSE and CBS were also identified immunohistochemically to be co-localized in the hypothalamic SON and PVN regions in magnocellular neurons, parenchyma, arteries and microvasculature (see Figure 2A). Hemorrhagic shock has been shown to induce hypothalamic OT release [28] and H_2_S can also stimulate hypothalamic OT release [4]. Thus, the colocalization of CSE, CBS, OT and OTR in the hypothalamus may be indicative of H_2_S stimulating the release of OT as a consequence of the dramatic fluid shifts generated by the hemorrhagic shock (see Figure 2B). Finally, it is tempting to speculate that endogenous H_2_S is involved in the hemorrhagic shock-induced OT release, although it is not yet clear how hemorrhagic shock affects the cerebral expression levels of the H_2_S-producing enzymes (CSE and CBS). Recently, in a porcine model of acute subdural hematoma-induced acute brain injury with resuscitation and neuro intensive care maintenance, Denoix et al. characterized the spatial expression pattern for the OT/OTR and the endogenous H_2_S-producing enzymes [25]. The authors found OT/OTR, CSE and CBS to be present in neurons, the vasculature and the parenchyma at the base of sulci. This finding might assume particular importance for translational purposes, because the base of the sulci is exactly where pressure-induced edema formation is reported to be found in the gyrencephalic human brain, which is not possible to investigate in the lissencephalic rodent brain [25]. Interestingly, the parenchyma and the cortical neurons in the gyri were positive for CSE but its expression was reduced with injury [25]. CBS, OT and OTR displayed an opposite pattern of expression to CSE and were upregulated with injury. The differential regulation of these enzymes suggests a much more complex relationship between the OT and H_2_S systems in osmotic regulation than what was reported previously [4,24].

Furthermore, H_2_S has been reported to improve the integrity of the barrier and attenuate cerebral edema, and the reduction of CSE expression close to injury sites coincided with increased Alb extravasation and barrier dysfunction [29,30,31]. Denoix et al. reported that OTR and CSE were also expressed in the arteries and microvasculature, suggesting that they might play a role in blood–brain barrier integrity [25]. The presence of OT/OTR in areas of acute subdural hematoma-induced injury reported by Denoix et al. is in line with OTR upregulation observed in humans, as an adaptive stress response in “vascular profiles” associated with perivascular swelling and around micro-infarcts [25,32].

## 3. H_2_S in Cardiac and Vascular Protection

Recently, in a mouse model of acute-on-chronic disease, Merz et al. showed that for mice with homozygous global deletion of CSE (CSEko) (generated by [17]), the main H_2_S producing enzyme in the cardiovascular system, chronically exposed to cigarette smoke prior to undergoing acute thorax trauma, displayed decreased OTR expression in the heart, which was attenuated by the administration of the slow-H_2_S-releasing compound GYY4137 [19]. Furthermore, these CSEko mice had significantly higher heart rates and blood pressure than wild type (WT) animals, and reduced OTR expression (see Figure 3A), supporting the important role of CSE expression in the cardiovascular system [16,17,19].

There are a plethora of fairly recent reviews on H_2_S and its protective effects in the cardiovascular system [7,33,34,35,36,37,38,39]. Endogenous H_2_S production or expression of H_2_S-producing enzymes (primarily CSE) has been reported in the following cell types of the cardiovascular system: smooth muscle cells, cardiomyocytes, endothelial cells and immune cells [8,19,25,40,41]. H_2_S has been shown to play a decisive role in the modulation of the cardiovascular system, i.e., as an endogenous activator of angiogenesis [8,42] via hypothalamic control [4], and a basal vasorelaxant, blood pressure and heart rate regulator [43,44]. Downstream signaling cascades involved in mediating H_2_S-dependent vaso-active effects activate K_ATP_-channels and stimulate Akt-dependent endothelial eNOS. Furthermore, H_2_S can induce antioxidant effects via the transcription factor nuclear factor erythroid 2-related factor 2 (Nrf2), which can be also regulated by OT [7,45,46,47]. H_2_S administration in rodents ameliorated myocardial fibrosis and oxidative stress in hypertension [48], and had beneficial effects on a myocardial infarct [15,49,50,51,52]. NaHS administration in rats significantly attenuated hemorrhagic shock-induced metabolic acidosis and simultaneously downregulated inducible nitric oxide synthase (iNOS) expression and NO production in the heart and aorta [53]. A murine model of combined blunt chest trauma and subsequent hemorrhagic shock showed that administration of the slow-releasing mitochondria targeted H_2_S donor AP39 during the resuscitation period and reduced lung tissue iNOS expression; however, it aggravated circulatory shock-induced hypotension due to its vasodilator capacities [54]. The complex interaction of H_2_S and NO in inflammation has been reviewed previously [55]. The H_2_S and OT systems share downstream signaling cascades that converge on the same NOS/NO-dependent pathway, which is further support of their interaction/relationship [7,56].

In large animal models, in general, the effects of H_2_S administration have been less robust than those seen in the rodent models. Sulfide administration in long-term porcine hemorrhage and resuscitation reduced mortality and attenuated organ dysfunction and injury, but its effectiveness in this model was restricted to a narrow timing and dosing window [57]. Other authors showed no benefit at all: intravenous Na_2_S did not induce hypothermia or improve survival from hemorrhagic shock in pigs [58,59]. Thus, for the clinical development of H_2_S-based therapies, which might potentially also affect OT/OTR signaling, further research is warranted.

## 4. OT/OTR in Cardiac and Vascular Protection

In contrast to the highly-diffusible, gaseous H_2_S, which does not require any membrane receptor [60], the OT system comprises a ligand–receptor (G protein-coupled) interaction. Upon OT binding to OTR, it can stimulate pro-survival kinases such as ERK and PI3K/Akt, which can in turn activate eNOS or CSE (H_2_S) [56]. OT can, furthermore, signal through calmodulin-dependent protein kinase II (CaMK II), regulating Ca^2+^ homeostasis, necessary for cardiomyocyte function [56]. OTR expression has been detected in cardiomyocytes, vasculature (smooth muscle cells and endothelium), macrophages, peripheral blood mononuclear cells and cardiac fibroblasts [19,61,62,63,64,65,66,67]. The role of OT in the heart has been recently reviewed [61,68], and interestingly enough OT shares many of the properties also reported for H_2_S, e.g., increase of glucose uptake in cardiac cells, anti-inflammatory and antioxidant activity [69,70,71], blood pressure lowering capacities via NO-mediated vasodilation [72], negative inotropic and chronotropic effects, natriuretic effects and effects on endothelial cell growth [14,68,73,74]. The NO-mediated vasodilatory effects of OT are also reported to regulate blood pressure [68,75,76,77,78] and body fluid homeostasis, albeit through an interaction with H_2_S [4,24].

Subcutaneous OT administration in myocardial infarct resulted in reduced inflammation, apoptosis, and ultimately ameliorated heart function [79,80,81,82]. Finally, downregulation of the OT system is associated with dilated cardiomyopathy [83], hypertension [84] and impaired cardiovascular function [19,46,68,80]. In a porcine myocardial infarct model, placebo-treated animals with high endogenous OT levels at the start of the experiment had better ejection fraction overall compared to OT-treated animals with high basal endogenous OT levels [85]. In contrast, in the low endogenous OT group receiving exogenous OT had no effect on cardiac function or cardiac OTR expression [85], yet the group with low basal OT levels without OT treatment had reduced function and larger infarct size [85]. Thus, the potential of OT to exert its cardio-protective effects seems to be at least in part dependent on the presence of both its basal receptor and its ligand.

## 5. Chronic Cardiovascular Disease

Administrations of both H_2_S [45,86,87] and OT [88,89] have been shown to reduce atherosclerotic plaque formation and to decrease the pro-inflammatory response—OT specifically through the upregulation of its receptor. Impaired endogenous H_2_S release is reported to be a key mediator with regard to the development of chronic cardiovascular pathology [8]. In a clinically relevant resuscitated model of septic shock, familial hypercholesterolemia Bretoncelles Meishan (FBM) swine, a comorbid large animal strain characterized by a clinical phenotype resembling the human atherosclerotic patient [90,91] with coronary artery disease, presented with elevated nitrotyrosine formation, a marker of nitrosative and oxidative stress, and significantly lowered CSE and OTR protein expression levels in the myocardium, which coincided with altered cardiac function [41,46]. Interestingly, already without septic shock, the FBM pig strain displayed decreased CSE expression in the media of the coronary artery and elevated nitrotyrosine formation [40]. The septic shock was associated with an even more pronounced downregulation of CSE [40]. Overall, these observations agree well with the fact that atherosclerosis and hypertension are associated with reduced levels of CSE [86]. Finally, CSE is proposed to modulate OTR in a tissue and function dependent manner during the progression of atherosclerosis [56]. The RISK pathway has been suggested to be the downstream molecular pathway, where H_2_S and OT signaling converge in atherosclerosis and cardioprotection (see Figure 3B, [56]). RISK activation leads to PI3K Akt, eNOS cascades and ERK 1/2 activation [56], which in turn promotes reperfusion by stimulating cell migration and angiogenesis. The PI3K/Akt cascades are also activated through H_2_S and are reported to promote myocardial protection [92]. These signaling pathways for H_2_S-mediated and OT-mediated myocardial protection were identified in animal models [56,92], but might also be relevant in human cardioprotection. Even though studies in human myocardial tissue are still lacking, the fact that H_2_S and OT activate the same downstream targets in human myometrial samples [10,11] supports this hypothesis. The RISK pathway is active in endothelial cells and cardiomyocytes. In endothelial cells, the activation of eNOS/NO as an angiogenic and vasodilating factor is crucial. In other cells, such as cardiomyocytes, RISK-activated pathways regulating apoptosis and antioxidant signaling play a role.

Interestingly, insulin receptors also act via the PI3K/Akt signaling pathway [93], and both OT and H_2_S are involved in the modulation of energy homeostasis and glucose metabolism (reviewed recently for OT [94] and H_2_S [95]). OT, CSE and CBS are expressed in insulin-sensitive tissues [95,96]. H_2_S is endogenously released by skeletal muscle, liver, adipose and islet-β-cells [97]; the OT and OTR are also present in islet-β-cells and skeletal muscle cells [98]. OT has been reported to increase glucose uptake in cardiomyocytes via PI3K signaling [96]; exogenous H_2_S administration restored blood glucose levels in the previously mentioned CSEko mice, and was associated with an upregulation of cardiac OTR [19]. Hyperglycemia, in turn, can reduce the expression of CSE [99] and suppress OT [68,100]. Low levels of OT and H_2_S are associated with diabetes and insulin resistance [68,95]. It is striking to observe that chronic co-morbidities in particular seem to have common ground in the interaction of H_2_S and OT.

## 6. The Role of OT/OTR in the Brain and Cardiovascular System during Psychological Stress

Psychological stress, e.g., early life stress, is well established as a risk factor for developing cardiovascular diseases [56,101,102,103,104]. The OT/OTR system is associated with stress-related responses, anxiolytic effects, maternal behavior, optimistic-belief updating, optimism and social reward perception, and it has been extensively studied in psychosomatic medicine [66,105,106]. Psychological stress was shown to increase heart rate and blood pressure, and the chemical blockade of the OTR worsens the cardiovascular response to stress [107,108,109,110]. An intracerebral injection of OTR antagonist attenuated the increase in heart rate after stress, but had no effect in the basal state without stress [110,111]. This suggests that multiple factors may be at play in the regulation of the OT system, stress being one of them, affecting endogenous receptor ligand levels and ultimately the physiological response.

OT plasma levels were positively related to improved cardiovascular and sympathetic responses to stress [112]. In mice, early life stress led to a significant long-term reduction of both cardiac CSE and OTR expression to the same degree [113]. Moreover, OTR knock out (OTRko) mice presented with reduced cardiac CSE expression, providing another confirmation of the link between the OT and H_2_S systems [113]. Interestingly, OTR expression was dependent on the “stress-dose”, inasmuch as “long-term” stressed (LTSS) mice had decreased OTR protein expression, whereas the “short-term” stressed (STSS) group displayed increased OTR expression [113]. This led the authors to conclude that the LTSS group with reduced OTR expression reflects increased vulnerability, whereas the STSS with higher receptor expression may indicate an adaptive response conferring resilience [113].

This provides evidence that the type of stress has an important impact on the OT ligand and its receptor modulation. In addition to the fact that the loss of trauma-related OTR in cardiac tissue was attenuated with administration of exogenous H_2_S [19], the interaction of the OT/OTR and the H_2_S/CSE systems in the context of psychological trauma was highlighted by the protective effect of exogenous H_2_S on early life stress-related colonic inflammation [114]. In turn, OT administration also exerted colon-protective effects through similar anti-oxidative and anti-inflammatory properties [115]. The therapeutic potentials of H_2_S and OT are beyond the scope of this work, but have been reviewed recently [56,116,117,118].

## 7. Conclusions

When reviewing the literature, it is striking that there is a disparity of studies of the H_2_S and OT system: there are plenty of reports on the role of H_2_S in physical trauma and OT in psychological trauma, whereas investigations of the role of H_2_S in psychological trauma and OT in physical trauma are rather limited. The current knowledge is not sufficient to speculate about treatment options; thus, this imbalance in studies should be addressed by researchers in the future. H_2_S and OT have parallel effects suggesting an interaction of the two systems. Both the H_2_S/CSE and OT/OTR systems assume major importance in the regulation of blood pressure and circulating blood volume. Moreover, there is evidence for their interaction with one another in peripheral organs, particularly in the heart. Finally, the two systems share signaling cascades that converge on the same signaling pathway. The interaction of the two systems seems to regard both psychological disorders and cardiovascular disease, and, hence, understanding more of the way that the H_2_S/CSE and OT/OTR systems work as mediators in stress may contribute to tackling the mutual interplay between body and mind.

## Figures and Tables

**Figure 1 antioxidants-09-00748-f001:**
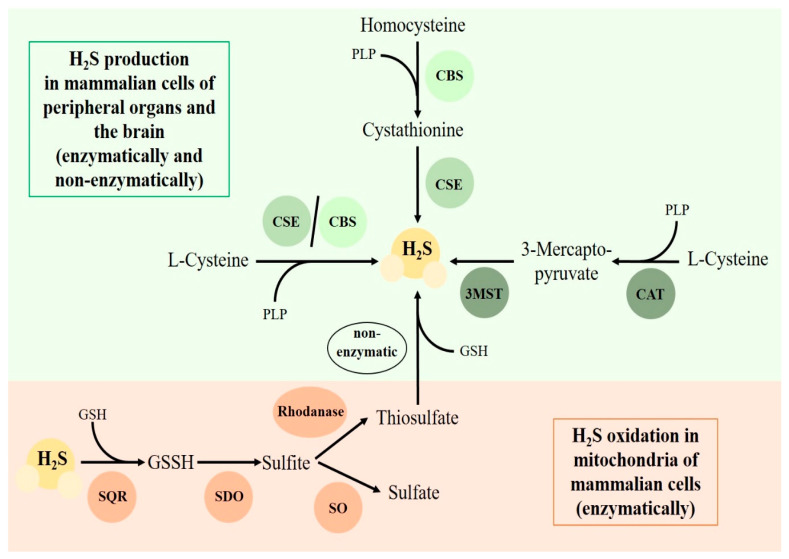
H_2_S biosynthesis and oxidation pathways. H_2_S can be produced in the vasculature, heart, brain, placenta, colonic tissue, liver, kidney and other mammalian tissues [8]. L-cysteine and homocysteine are the essential substrates for enzymatic endogenous H_2_S production by CSE, CBS or 3MST. L-Cysteine can be directly used as a substrate for H_2_S generation via the enzymes CSE or CBS. Homocysteine is converted to cystathionine by CBS and then converted to H_2_S by CSE. Cysteine-aminotransferase (CAT) can metabolize L-cysteine to 3-mercaptopyruvate, which is then converted to H_2_S by 3MST. CSE, CBS and CAT require the cofactor pyridoxal 5′-phosphate (PLP) [8]. Non-enzymatic H_2_S generation can take place during hypoxic events, for example, via thiosulfate utilization. Thiosulfate is an oxidation product of H_2_S, which is part of the stepwise enzymatic sulfide oxidation pathway within the mitochondria and can be reduced, e.g., with glutathione (GSH), to H_2_S [8,9]. Sulfide quinone oxidoreductase (SQR) oxidizes H_2_S to glutathione persulfide (GSSH); that is followed by oxidation of GSSH to sulfite by persulfide dioxygenase (SDO); and finally, sulfite is oxidized to either sulfate by sulfite oxidase (SO) or to thiosulfate by rhodanese.

**Figure 2 antioxidants-09-00748-f002:**
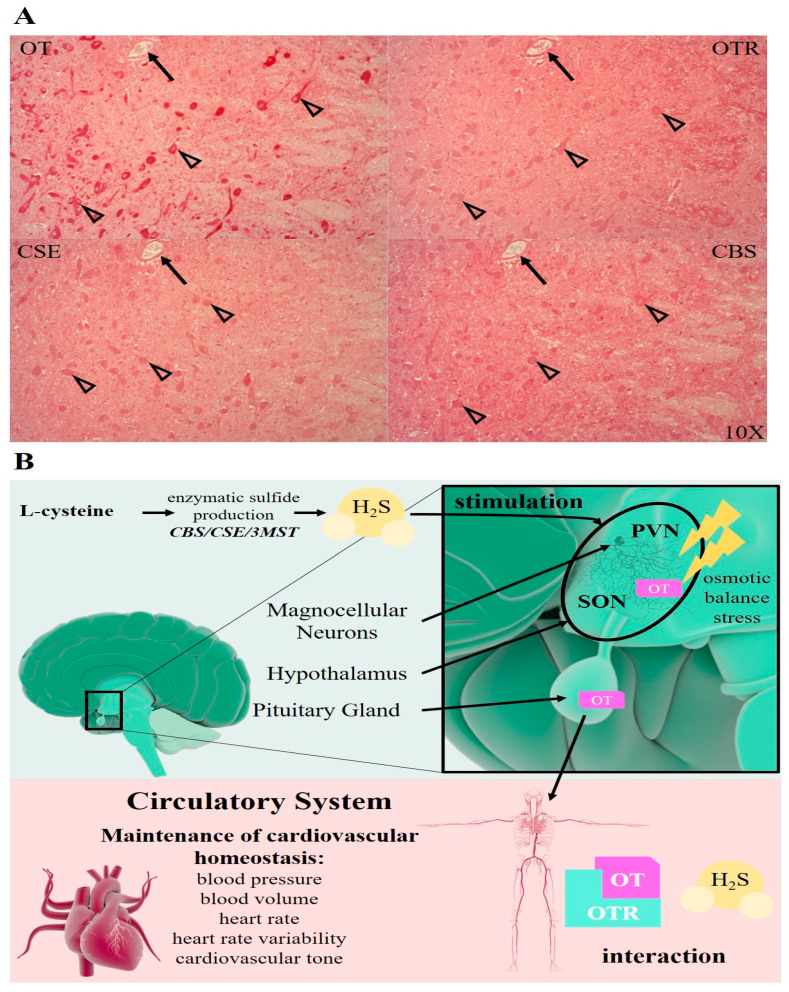
H_2_S can stimulate hypothalamic OT release. (**A**): OT, OTR, CSE and CBS co-localize (in red) in the magnocellular neurons (open arrow heads) of the porcine hypothalamus. Black arrows are pointing to the same blood vessel in consecutive brain sections of the porcine hypothalamus (10×). Brain sections were obtained from an atherosclerotic pig model of resuscitated hemorrhagic shock [27]. Hypothalamic immunohistochemical pictures have been previously published in an abstract in: 39th International Symposium on Intensive Care and Emergency Medicine. Crit Care 23, 72 (2019). https://doi.org/10.1186/s13054-019-2358-0 (http://creativecommons.org/licenses/by/4.0/). The used antibodies have been verified for their specificity previously by Denoix et al. [25]. (**B**): H_2_S can stimulate OT release in the hypothalamus, including the PVN and SON, in response to osmotic balance stress and fluid shifts. H_2_S is endogenously enzymatically produced by CSE, CBS and 3MST. Hypothalamic OT is released via the posterior pituitary into the circulation, where it contributes to the maintenance of cardiovascular homeostasis. Illustrations of the heart, brain, neurons and the circulatory system were taken from the Library of Science and Medical lllustrations (somersault18:24, https://creativecommons.org/licenses/by-nc-sa/4.0/).

**Figure 3 antioxidants-09-00748-f003:**
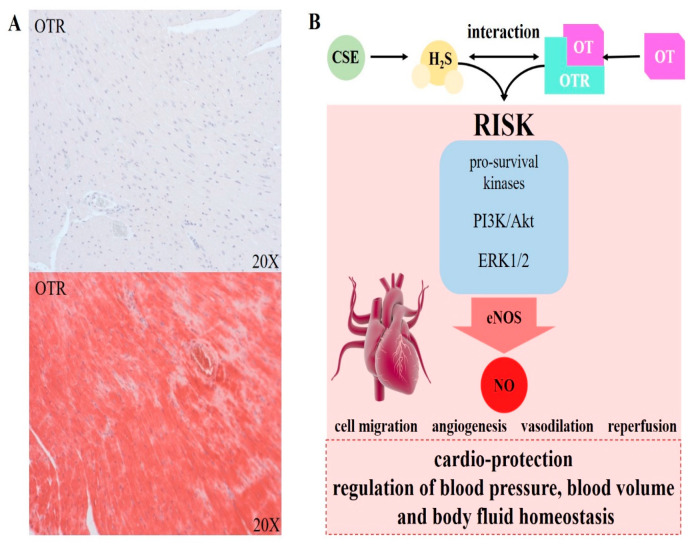
Interaction of H_2_S and OT in the heart. (**A**): Immunohistochemical staining of OTR in heart tissue in a CSEko heart (*n* = 3, top) and a wild-type heart (*n* = 3, bottom). Heart samples were obtained from naïve animals, which were anesthetized with sevoflurane and buprenorphine and sacrificed via exsanguination (as previously described in [19]). Expression of OTR was absent in CSEko heart and clearly visible in wild-type myocardial tissue. (**B**): The physiological basis of the interaction of CSE and the OTR, converging in the reperfusion injury salvage kinase (RISK) pathway. H_2_S, mainly produced by CSE in the cardiovascular system, and the OTR, stimulated by OT, can both activate the RISK pathway. RISK activation leads to PI3K, protein kinase B (Akt), ERK1/2 cascades and endothelial nitric oxide synthase (eNOS) activation, and promotes reperfusion by stimulating cell migration, angiogenesis and vasodilation, resulting in cardio-protection, regulation of blood pressure, blood volume and body fluid homeostasis. The illustration of the heart was taken from the Library of Science and Medical Illustrations (somersault18:24, https://creativecommons.org/licenses/by-nc-sa/4.0/).

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
