# Peer review of "The Interaction of the Endogenous Hydrogen Sulfide and Oxytocin Systems in Fluid Regulation and the Cardiovascular System"

_antioxidants, 2020, doi:10.3390/antiox9080748_

Round 1

Reviewer 1 Report

In this review manuscript, Denoix et al., reviewed the hydrogen sulfide (H2S) and oxytocin (OT) in cardiovascular regulation and fluid homeostasis. Interaction between these two pathways is recently studied. It seems that this manuscript timely wraps up the recent findings, give a good introduction into this interesting research area. However, more backgrounds and information should be required to improve this review paper.

Major

1. Hydrogen sulfide biogenesis

The authors should explain more about hydrogen sulfide biogenesis. The authors briefly mentioned enzymes in the introduction, but not any further details. Because many readers are not familiar with this pathway, more detailed explanations about hydrogen sulfide are necessary. For example, what is a substrate? Which enzymes are involved in biosynthesis and which step (CSE and CBS)? Which tissues are producing this? What are the physiological roles of hydrogen sulfide? It would be great if the authors provide an illustration. 

2. Neuroendocarine control of fluid hemostasis and Figure 1

Why the hypothalamus are of particular interest? Please demonstrate the roles of the hypothalamus. 

Figure 1B should be revised. It is unclear where is the hypothalamus in the brain and where are PVN and SON in the hypothalamus region. Please revise a figure, in which please draw a more detailed anatomy of the hypothalamus (along with pituitary gland) PVN, SON and magnocellular neurons.  

3. The cellular sources of H2S and OT/OTR in hearts and cardiovascular systems. 

Which cells are responsible for H2S and OT/OTR? Do cardiomyocytes express OTR? Do endothelial cells are responsible to sense H2S? If not known, please indicate it. Which cells are responsible for the RISK activation, downstream eNO cascades? Do endothelial cells secrete NO to promote the migration of other endothelial cells? Please clarify these. 

4. Line 120-121

CSEko mice: please clarify whether this is global CSE knockout mice or conditional knockout one.

Reviewer 2 Report

The review manuscript by Denoix et al discussed the interaction of H2S and OT systems in the cardiovascular system. This is an important topic to be covered. My suggestions for the authors how to improve their manuscript are below. 

  1. Some physiological background in HTS and OT/OTR signaling would be helpful in the introduction.
  2. The text a little bit seems to be a compilation of details from the literature with not too much discussion from the authors, which would significantly enrich the review. The manuscript would benefit from a more insightful discussion of the literature and the authors personal view on H2S and OT signaling at the end of each section.
  3. 1A. Is it a published abstract? If yes, was permission obtained from the publisher to use this and other figures/illustrations shown in the manuscript? It is unclear how OT, OTR, CSE and CBS expression can be compared using IHC as staining intensity may vary independent of differences in protein expression.
  4. The authors propose that H2S release is increased by hemorrhagic shock but no tissue staining in control animals has been provided.
  5. No methodological information and number of independent experiments have been disclosed in Fig. 2A legends, as such these experiments cannot be performed by others and it is not known if they are reproducible. It is recommended to discuss only published results and remove unpublished data from the manuscript.
  6. CSE is described as the main producer of H2S in the cardiovascular system and OTR activates RISK, leading to eNOS activation. What is known about the expression of H2S producing enzymes and OTR in other cell types relevant to cardiovascular pathology (SMC, fibroblasts, and inflammatory cells)?
  7. H2S and OTR signaling activates PI3K/Akt in endothelial cells. Does H2S and OTR interfere with insulin receptor signaling in skeletal muscle, adipocytes and liver? Any roles in metabolism or insulin resistance?
  8. H2S and OTR signaling in human cells and tissue has not been described and translational importance remains unclear.
  9. The review in general should include more details. Results sometimes are discussed in a way that no major conclusions can be drawn. For example,

“Inadequate endogenous H2S release is reported to be a key mediator with regards to the development of chronic cardiovascular pathology.”

“An intracerebral injection of OTR antagonist improved the cardiovascular function after stress, but had no effect in the basal state without stress [93,94]”

“In mice, early life stress led to a long term reduction of both cardiac CSE and OTR expression, which were directly related to one another”. The goal of this review is  to describe the interaction between H2S and OT. The results that suggests the interaction should be more detailed.

  1. Future directions and possible therapeutic interventions targeting H2S and OT in physical and psychological trauma should be included in the manuscript.

Round 2

Reviewer 1 Report

Thank you for the authors to address my comments.

I don't have any more comments.

Author Response

Thank you.

Reviewer 2 Report

My comments have been addressed. 

Author Response

Thank you.